# Histone Acetylation Dynamics during In Vivo and In Vitro Oocyte Aging in Common Carp *Cyprinus carpio*

**DOI:** 10.3390/ijms22116036

**Published:** 2021-06-03

**Authors:** Swapnil Gorakh Waghmare, Azin Mohagheghi Samarin, Azadeh Mohagheghi Samarin, Marianne Danielsen, Hanne Søndergård Møller, Tomáš Policar, Otomar Linhart, Trine Kastrup Dalsgaard

**Affiliations:** 1South Bohemian Research Center of Aquaculture and Biodiversity of Hydrocenoses, Research Institute of Fish Culture and Hydrobiology, Faculty of Fisheries and Protection of Waters, University of South Bohemia in Ceske Budejovice, 389 25 Vodňany, Czech Republic; mohagheghi@frov.jcu.cz (A.M.S.); amohagheghi@frov.jcu.cz (A.M.S.); policar@frov.jcu.cz (T.P.); linhart@frov.jcu.cz (O.L.); 2Department of Food Science, Aarhus University, Agro Food Park 48, 8200 Aarhus, Denmark; marianne.danielsen@food.au.dk (M.D.); hsm@food.au.dk (H.S.M.); trine.dalsgaard@food.au.dk (T.K.D.); 3Center of Innovative Food Research, Aarhus University Centre for Innovative Food Research, 8000 Aarhus, Denmark; 4CBIO, Aarhus University Centre for Circular Bioeconomy, 8000 Aarhus, Denmark

**Keywords:** *Cyprinus carpio*, egg quality, epigenetics, histone acetyltransferase, histone modifications, post-ovulatory aging

## Abstract

Aging is the most critical factor that influences the quality of post-ovulatory oocytes. Age-related molecular pathways remain poorly understood in fish oocytes. In this study, we examined the effect of oocyte aging on specific histone acetylation in common carp *Cyprinus carpio*. The capacity to progress to the larval stage in oocytes that were aged for 28 h in vivo and in vitro was evaluated. Global histone modifications and specific histone acetylation (H3K9ac, H3K14ac, H4K5ac, H4K8ac, H4K12ac, and H4K16ac) were investigated during oocyte aging. Furthermore, the activity of histone acetyltransferase (HAT) was assessed in fresh and aged oocytes. Global histone modifications did not exhibit significant alterations during 8 h of oocyte aging. Among the selected modifications, H4K12ac increased significantly at 28 h post-stripping (HPS). Although not significantly different, HAT activity exhibited an upward trend during oocyte aging. Results of our current study indicate that aging of common carp oocytes for 12 h results in complete loss of egg viability rates without any consequence in global and specific histone modifications. However, aging oocytes for 28 h led to increased H4K12ac. Thus, histone acetylation modification as a crucial epigenetic mediator may be associated with age-related defects, particularly in oocytes of a more advanced age.

## 1. Introduction

High-quality eggs are essential for proper embryo development and good health in the later life of the offspring [1]. Internal components such as maternal proteins or gene transcripts and external factors such as brood stock diet, environmental conditions, husbandry practices, and oocyte aging can influence the quality of oocytes [2]. The age of the oocyte has been recognised as a factor that affects egg quality after ovulation [3,4]. It has been observed that genetic and epigenetic changes within the genome can affect the developmental competence of the eggs, and these changes can be inherited by the offspring [5,6]. Fertilisation occurs within a short time after the release of the metaphase II oocyte from the follicle at ovulation [7]. Delays in fertilisation can result in post-ovulatory oocyte aging. In fish, this type of oocyte aging can occur due to defects in egg spawning or stripping.

Aging is a multifaceted process that is characterised by genetic and epigenetic changes within the genome and that involves various molecular pathways [8]. To date, only a small number of reports have described the molecular changes that occur during fish oocyte aging [9,10,11,12,13,14]. Epigenetics, the link between the environment and genes, has been suggested as a likely contributor to the aging phenotype [15,16]. Different epigenetic regulators/marks such as DNA methylation, histone modifications, and non-coding RNAs are associated with the aging process in other organisms. Previous studies on oocyte aging indicated the DNA methylation changes in bovine [17], histone modifications in mice and porcine [18,19], and modified expression of non-coding RNAs in rainbow trout (*Oncorhynchus mykiss*) [12]. The epigenome of an organism is extensively reprogrammed during gametogenesis and early embryo development through these epigenetic regulators/marks. Furthermore, post-fertilization success and healthy offspring development rely on epigenetic reprogramming [20]. Similarly, epigenetic alterations may be responsible for age-related complications in fish oocytes.

Histone modifications are one of the epigenetic mechanisms that have been suggested to be involved in the oocyte aging process [15,16]. Histones are the most abundant protein components within the chromatin structure, where they form nucleoprotein complexes. They are the building blocks of the eukaryotic chromatin structure and are extensively conserved during evolution [21]. The reactions/processes associated with DNA occur at the nucleoprotein complex. A nucleosome is the basic unit of chromatin structure and consists of four different histones (H2A, H2B, H3, and H4) in the form of dimers that are collectively arranged as octamers [22]. The nucleosome possesses an N-terminal tail that is rich in arginine and lysine, and this is the site for post-translational modifications [23]. Histone modifications consist of different types that include acetylation and methylation of lysines and arginines, phosphorylation of serines and threonines, ubiquitylation, and ribosylation of lysines. Histone acetylation occurs more frequently than does any other modification [24,25]. Histone acetyltransferases (HATs) catalyse histone acetylation by transferring acetyl groups from acetyl coenzyme A (acetyl-CoA) to lysine residues within the core histones. Conversely, histone deacetylases (HDACs) catalyse the removal of acetyl groups. Together, HATs and HDACs regulate the acetylation status of histones. Histone acetylation and deacetylation play critical roles in controlling gene expression and several cellular functions [26,27]. Increased histone acetylation levels and histone acetyltransferase 1 (*hat1*) transcripts were observed during prolonged oocyte aging in mice [19], porcine [18], and goat [28]. The underlying pathways responsible for these alterations during fish oocyte aging remain unclear.

Fish exhibit benefits over other vertebrates in regard to the study of oocyte aging. The high number of oocytes produced by female fish and the diversity of reproductive modes make it possible to properly study the oocyte aging process. Specifically, studying oocyte aging at the molecular level may aid aquaculture and fisheries in managing egg quality. The current study employed Western blotting, supporting the reports of histone modifications during oocyte aging using immunohistochemistry in other vertebrates. This study was performed on common carp (*Cyprinus carpio*) because of synchronous ovulation in females and based on our previous satisfactory experience with the practical approaches. Additionally, common carp is one of the major aquaculture species in which the quality of eggs has been relatively well documented [29] and there is no need to sacrifice the experimental animal as it might be required in other vertebrates.

Histone modification dynamics were investigated during both in vivo and in vitro oocyte aging in common carp. Acetylation modifications at lysines 9 and 14 on histone H3 and at lysines 5, 8, 12, and 16 on histone H4 were analysed using specific antibodies. Additionally, a histone acetyltransferase activity assay was used to determine HAT activity in fresh and aged oocytes both in vivo and in vitro. The obtained results are the first to report histone modifications as an important epigenetic regulator during oocyte aging in fish and may also aid in the further development of assisted reproduction technology in higher vertebrates.

## 2. Results

### 2.1. Egg Quality Indices

The egg eyeing and hatching rates were significantly affected by post-ovulation and post-stripping oocyte aging, as presented in Table 1.

### 2.2. Image Analysis of 2D AUT × SDS PAGE

In total, 37 protein spots matched among freshly ovulated and in vivo and in vitro aged oocyte samples (Appendix A). The image analysis of 2D AUT × SDS PAGE revealed no significant differences between freshly ovulated oocytes and either in vivo or in vitro aged oocytes. However, at low significance (*p* < 0.1), spot numbers 80 (in vivo) and 113 (in vitro) were significantly different compared to those of the freshly ovulated oocyte samples. Spot number 80 exhibited a 1.3-fold decrease, while spot number 113 showed a significant 1.6-fold increase in oocyte aging.

### 2.3. Histone Acetylation during In Vivo and In Vitro Oocyte Aging 

The acetylation of histones at H3K9, H4K5, and H4K8 did not significantly change during in vivo or in vitro oocyte aging (Figure 1A,B,D). Acetylation of H4K12 increased significantly in vitro but not during in vivo oocyte aging (Figure 1B). A comparison of histone acetylation at 8 and 28 h in vivo and in vitro did not reveal any significant changes (Figure 1C). Acetylation at H3K14 and H4K16 did not exhibit any signal in either freshly ovulated or in vivo and in vitro aged oocytes (Figure 1E). The efficiency of the related antibodies was confirmed by the presence of both modifications expressed in the mouse liver used as the positive control.

### 2.4. Histone Acetyltransferase Activity during In Vivo and In Vitro Oocyte Aging 

Histone acetyltransferase activity was measured in freshly ovulated and in 8 h and 28 h in vivo and in vitro aged oocytes. HAT activity did not change significantly during in vivo or in vitro oocyte aging; however, a slight increasing trend was observed (Figure 2). A comparison of histone acetyltransferase activity at 8 and 28 h in vivo and in vitro also did not reveal any significant changes.

## 3. Discussion

The competence of the oocyte to fertilise and develop into a normal embryo declines with increased post-ovulatory oocyte age that begins at ovulation and progresses constantly until fertilisation [1]. The mechanisms driving the oocyte aging process are not yet clear. Histone modifications are the most common and important epigenetic configuration that may contribute to age-associated defects in oocytes and the arising embryos. Establishment of chromatin structures is highly controlled by histone proteins. Many initial and essential functions, including cell cycle progression, DNA replication and repair, transcriptional activity, and chromosome stability, are associated with histone modifications [30,31,32,33].

In the present study, the results of global histone modification analysis suggest that no histone modifications are altered during post-ovulatory oocyte aging. However, at low significance (*p* < 0.1), spot numbers 80 (in vivo) (Appendix A) and 113 (in vitro) (Appendix A) were significantly different compared to those of the freshly ovulated oocyte samples. Spot number 80 exhibited a 1.3-fold decrease, while spot number 113 showed a significant 1.6-fold increase in oocyte aging. Post-translational modifications of H3 and H4 are much more widely identified than modifications on other histone proteins [34]. Acetylation modification of histones H3 and H4 has been reported in other vertebrates during post-ovulatory oocyte aging [15,16]. The key sites of lysine acetylation include two lysines on histone H3 (K9 and K14) and four lysines on histone H4 (K5, K8, K12, and K16), which are described as active transcription marks [34,35]. The acetylation and deacetylation of these N-terminal lysine residues both play a critical role in regulating chromatin condensation and folding, heterochromatin silencing, and transcription, and based on this they influence various cellular processes [27,36]. Therefore, we studied specific histone acetylations (H3K9ac, H3K14ac, H4K5ac, H4K8ac, H4K12ac, and H4K16ac) during in vivo and in vitro oocyte aging.

Among the selected histone acetylation modifications examined in this study, H3K9ac, H4K5ac, and H4K8ac exhibited no significant differences during in vivo and in vitro oocyte aging. However, Huang, et al. [19] reported no signal for histone acetylation at H3K9 and H4K5 and increased H4K8ac during in vivo and in vitro mouse oocyte aging. In contrast, Xing et al. [37] observed an increase in H3K9ac in aged mouse oocytes and attributed the difference to the different implemented oocyte culture media and antibodies. In our study, H3K14 and H4K16 were not acetylated in either fresh or aged common carp oocytes. Liu et al. [38] also did not detect a signal for H3K14ac in fresh oocytes; however, this signal was detected in aged mouse oocytes. A gradual increase in H3K14ac was reported during oocyte aging in mice [19]. In vivo and in vitro aged mouse oocytes displayed a signal for the acetylation of H3K14 in 76% and 75% of the oocytes, respectively, and no signal was observed in freshly ovulated oocytes [38]. All of the investigated histone acetylation modifications in our study have been reported to be species-specific and oocyte stage dependent [34]. Accordingly, the results of our study revealed undetected H3K14ac and H4K16ac in common carp MII oocytes. In future studies, it would be of interest to investigate the dynamics of different histone modifications during fish oocyte maturation and embryo development.

In the current study, the acetylation on H4K12 was increased significantly at 28 HPS. This is in accordance with previous findings in other vertebrates. Immunofluorescence detection of H4K12 acetylation in mouse oocytes revealed an increased signal during 5 and 10 h of aging [19]. All 12 h-aged mouse oocytes exhibited signals for the acetylation on H4K12, while no signal was detected in freshly ovulated oocytes [38]. Increased fluorescence signals have been reported for H4K12ac in 24 h-aged porcine and mouse oocytes [18,39]. Cui et al. [18] suggested that increased H4K12ac in 28 h-aged porcine oocytes is associated with increased oxidative stress within the ooplasm. The observed increase in H4K12ac levels in oocytes aged for 28 h post-stripping in the current study may therefore be a consequence of increased oxidative stress at this time point; however, a previous study examining common carp indicated that oxidative stress is not likely to be the main initiator of post-ovulatory aging in common carp oocytes, at least up to 14 h in vivo and 10 h in vitro [40]. Moreover, the observed late onset of hyperacetylation of H4K12 suggests the possible optimisation of egg storage in common carp. H4K12ac is essential for centromere protein A deposition into centromeres [41], a process that is required for accurate chromosome segregation during cell division [42]. Furthermore, H4K12ac has been suggested to play a critical role in loosening chromatin structures during DNA replication [43]. Post-ovulatory oocyte aging may be one of the factors that spontaneously release the zygotic clock to thereby trigger molecular pathways [44].

The zygotic clock is a molecular clock that initiates cascades of biochemical processes that occur post-fertilization or after egg activation [45]. Post-translational histone modifications are among the mechanisms underlying zygotic genome activation [46]. Histone acetylation is a marker of active transcription that is projected to increase during the maternal to zygotic transition and is linked to the activation of genes [46,47]. Therefore, the observed hyperacetylation of H4K12 in aged oocytes in this study may be due to the spontaneous activation and ticking of the zygotic clock. Some studies have also reported dynamic changes in transcripts and in protein abundance during post-ovulatory oocyte aging that are known to be transcriptionally inactive [13,28,39,48]. While post-ovulatory oocyte aging does not lead to complete zygotic genome activation, it may trigger some of the pathways and partially explain the functional consequences of oocyte aging. Until now, few studies have compared the oocyte aging in vivo and in vitro conditions. The comparison of histone acetylation at 8 and 28 h in this study did not reveal any significant changes between in vivo and in vitro aging. This is in accordance with the previous finding by Zhang et al. [49], who reported no significant difference in the decreased expression of selected histone deacetylases transcripts during 24 h in vivo and in vitro oocyte aging in mouse.

The acetylation–deacetylation switch depends upon different physiological conditions, and this switch is achieved through the action of HATs and HDACs [20]. The HAT activity in the current study tended to increase during oocyte aging, although the difference was not statistically significant. *Hat1* transcript levels have also been reported to increase during porcine and goat oocyte aging [18,28]. The mRNA levels of the histone deacetylases *sirt1*, *sirt2* and *sirt3* were downregulated in mouse oocytes that were aged in vivo and in vitro [49]. Another study reported the downregulation of a gene responsible for histone deacetylation (*hdac2*) during maternal oocyte aging in mice [50]. These altered expressions of key epigenetic modulators in aged oocytes may lead to abnormal histone acetylation.

## 4. Materials and Methods

The histone modification dynamics during in vivo and in vitro oocyte aging in common carp was assessed through three experiments as: (I) egg storage and quality assessment, (II) histone modifications and (III) histone acetyltransferase activity assay.

### 4.1. Egg Storage and Quality Assessment

The in vivo and in vitro egg storage and the embryo quality assessment were done as bellow:

#### 4.1.1. Experimental Fish Preparation 

The broodfish preparation and artificial reproduction processes were performed according to Samarin et al. [51] and Samarin et al. [40]. Briefly, the experimental fish were treated with a gradual increase in water temperature from 16 °C to 20 °C. The carp pituitary hormone was used to stimulate ovulation and spermiation according to the method of Horvath and Tamas [52]. The experiment was conducted using six females for both the in vivo and in vitro experiments. 

#### 4.1.2. In Vivo and In Vitro Oocyte Aging 

Oocytes from females were individually incubated in vivo for 28 h post-ovulation (HPO) and in vitro for 28 h post-stripping (HPS) at 20 °C. The stored oocytes in vivo were stripped and fertilised at the time of ovulation (0 HPO) and then at 2 h intervals up to 12 HPO and also at 28 HPO. The stored oocytes in vitro were fertilised at the time of stripping (0 HPS) and then at 2 h intervals up to 10 HPS and also at 28 HPS. The in vivo and in vitro egg storage conditions were determined according to Samarin et al. [40].

#### 4.1.3. Artificial Fertilisation and Egg Developmental Success 

Eyeing and hatching rates were recorded as the egg quality indices. Three days post-fertilisation, the ratio of the number of eyed eggs to the number of initial inseminated eggs was used to calculate eyeing percentages. Hatching rates were measured after six days of fertilisation using the number of hatched larvae to the number of initially inseminated eggs. All of the steps for artificial fertilisation, removing egg stickiness, incubation, and assessment of egg developmental success were performed according to Samarin et al. [51].

### 4.2. Histone Modifications

The global histone modifications and specfic histone acetylation were investigated as bellow:

#### 4.2.1. Sample Collection for Histone Modification Analysis

Approximately 1 g of oocytes was sampled from individual females separately at each HPO and HPS. The collected samples were placed into cryotubes (Thermo Fisher Scientific, Waltham, MA, USA), labelled, frozen in liquid nitrogen, and stored at −80 °C for further investigation.

#### 4.2.2. Histone Isolation

Histone isolation was performed for samples collected at 0, 8, and 28 HPO and HPS. The isolation procedure was performed according to Wu et al. [53] with some modifications. Briefly, 150 mg of frozen oocyte samples were resuspended and homogenised in a 1.5 mL tube containing 400 µL of homogenisation buffer (10 mM Tris-HCl; 1 mM MgCl_2_; pH 7.5) and 20 µL of 5% digitonin (D141, Sigma-Aldrich, St. Louis, MO, USA). After centrifugation, the supernatant was discarded, and 1 mL of salt wash buffer (10 mM Tris-HCl; 1 mM MgCl_2_; 0.4 M NaCl; pH 7.5) was added to the pellet and incubated on ice for 15 min. Samples were then centrifuged, and the pellets were used for acid extraction of histones. Acid-extracted histones were precipitated using saturated trichloroacetic acid (T6399, Sigma-Aldrich, St. Louis, MO, USA). The air-dried histone pellets were then dissolved in 50 µL of Milli-Q water and stored at −20 °C for further analysis. The quality and quantity of isolated histones were both assessed by separating 5 µL of the histone on 4–15% Criterion™ TGX™ Precast Midi Protein Gels (Bio-Rad, Hercules, CA, USA).

#### 4.2.3. Separation of Histone Modifications

The 2D AUT × SDS PAGE (acetic acid-urea-Triton sodium dodecyl sulfate polyacrylamide gel electrophoresis) assay was used to separate the isolated histones. The Mini Trans-Blot^®^ Cell system (Bio-Rad, Hercules, CA, USA) was used for both the first- and second-dimension separations. The gel electrophoresis procedure was adopted from Shechter et al. [54] and Green and Do [55]. In summary, histone samples were separated on 15% AUT-PAGE, and the running buffer (5% acetic acid) was used for electrophoresis at 200 V for 140 min. Thereafter, the AUT gel target lanes were cut individually and carefully placed on top of the SDS-PAGE gel (4–15%). The stacking gel was poured around the AUT gel lane and allowed to polymerise. The second-dimension gel was run at a constant voltage of 150 V for 90 min, and this was followed by Coomassie staining. Gel images were recorded using the ChemiDoc™ XRS^+^ system (Bio-Rad, Hercules, CA, USA). Image analysis of 2D AUT × SDS PAGE was performed using Prodigy SameSpots version 1.0 software (Nonlinear Dynamics, Newcastle, UK).

#### 4.2.4. Immunodetection of Specific Histone Acetylation

For the analysis of specific histone acetylation modifications, isolated histone samples were separated on 4–15% Criterion™ TGX™ Precast Midi Protein Gels (Bio-Rad, Hercules, CA, USA). The isolated histone proteins were transferred to a 0.2 μm nitrocellulose membrane using the Trans-Blot Turbo Transfer Pack (Bio-Rad, Hercules, CA, USA) and the Trans-Blot Turbo Blotting System (Bio-Rad, Hercules, CA, USA) at 2.5 A and 25 V for 7 min. The membranes were then blocked for 1 h at room temperature in 5% bovine serum albumin (BSA) in Tris-buffered saline-Tween 20 (TBST) buffer. Thereafter, the blot was incubated overnight at 4 °C with the primary antibody diluted in blocking solution according to the manufacturer’s instructions (Abcam, Cambridge, UK). The selected specific histone acetylation modifications were studied using the following antibodies: Anti-AcH3K9, ab10812; Anti-AcH3K14, ab52946; Anti-AcH4K5, ab51997; Anti-AcH4K8, ab15823; Anti-AcH4K12, ab46983; Anti-AcH4K16, ab109463 (Abcam, Cambridge, UK). Separate blots were used to analyse the modifications. After incubation with the primary antibody, the blots were washed in TBST buffer and incubated with the secondary antibody that consisted of goat anti-rabbit IgG HRP conjugate (12-348, Millipore, Billerica, MA, USA) diluted 1:8000 in 1% BSA prepared in TBST buffer for 1 h at room temperature. After washing in TBST, the blots were developed using an ECL kit (1705061, Bio-Rad Laboratories, Hercules, CA, USA).

### 4.3. Histone Acetyltransferase Activity Assay

HAT activity was measured using a Histone Acetyltransferase Activity Assay Kit (Colorimetric) (ab65352, Abcam, Cambridge, UK) according to the manufacturer’s instructions. Briefly, approximately 50 mg of the oocyte samples at 0, 8, and 28 HPO and HPS were homogenised in 0.1% Triton X-100 on ice and then centrifuged at 10,000× *g* for 5 min at 4 °C. The supernatant was used for histone acetyltransferase activity assays. The total protein concentration was estimated using the Bio-Rad protein assay kit (500-0001, Bio-Rad Laboratories, Hercules, CA, USA). Fifty micrograms of protein extract were incubated with HAT substrate I, HAT substrate II, and NADH-generating enzyme in HAT assay buffer for up to 4 h at 37 °C. An ELISA plate reader (Synergy 2, BioTek Instruments, Winooski, VT, USA) was used to measure the absorbance at 440 nm. The active nuclear extract was used as a positive control and a standard. All measurements were performed in duplicate.

### 4.4. Statistical Analysis

The 2D AUT × SDS PAGE images were normalised and analysed using the Prodigy SameSpots software (Nonlinear Dynamics, Newcastle, UK) to determine differences in staining intensities among gels. The protein spots exhibiting a >1.5 average fold difference in spot volume were recognised as differentially expressed between the fresh and aged oocyte samples. Western blot images were normalised using the corresponding stain-free gel image and quantified using Image Lab 6.1.0 (Bio-Rad Laboratories, Hercules, CA, USA) software. The quantified Western blot images and histone acetyltransferase activity data were analysed using GraphPad Prism 9.1.0 (GraphPad Software, San Diego, CA, USA). Tukey’s multiple comparisons test was used to determine the differences in histone acetylation and histone acetyltransferase activity assays during oocyte aging. Additionally, histone acetylation and histone acetyltransferase activity were compared between oocytes aged 8 and 28 h in vivo and in vitro. Differences were considered significant at *p* < 0.05.

## 5. Conclusions

This study confirmed the presence of acetylation markers on H3K9, H4K5, H4K8, and H4K12 in common carp metaphase II oocytes. There was no evidence of acetylation at H3K14 and H4K16 in either fresh or aged oocytes. Furthermore, an increased acetylation pattern of H4K12 was observed during 28 h of in vitro oocyte aging in common carp. These findings highlight the dynamics of other histone modifications and epigenetic regulators during fish oocyte aging. Furthermore, it will be of interest to investigate the genomic regions that are associated with the hyperacetylation that occurs due to fish oocyte aging.

## Figures and Tables

**Figure 1 ijms-22-06036-f001:**
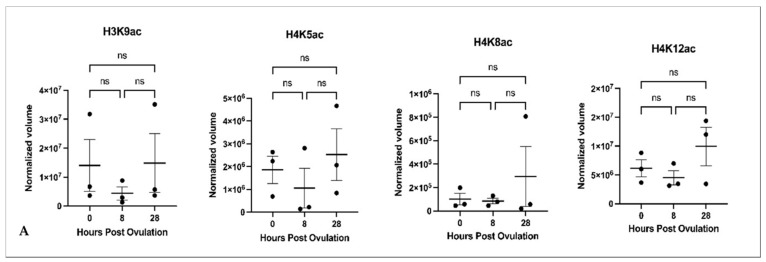
Effect of in vivo (**A**) and in vitro (**B**) oocyte aging on the acetylation of the selected histone lysines in common carp (*p* < 0.05, Tukey’s multiple comparisons test; ns: non-significant; ** *p* < 0.01, mean ± SEM); (**C**) Comparison of histone acetylation at 8 and 28 h between in vivo and in vitro aging conditions. (**D**) Western blot images for H3K9ac, H4K5ac, H4K8ac and H4K12ac; (**E**) Western blot images for H3K16ac and H4K514ac including positive control (mouse liver histones). The numbers in red colour indicate the biological replicates as: fish 1 (1, 4, 7, 10), fish 2 (2, 5, 8, 11, 13), and fish 3 (3, 6, 9, 12, 14).

**Figure 2 ijms-22-06036-f002:**
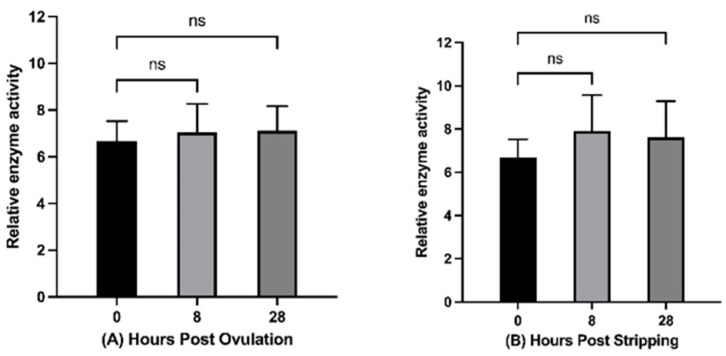
Effect of in vivo (**A**) and in vitro (**B**) oocyte aging at 20 °C on the histone acetyltransferase activity in common carp (*p* < 0.05, Tukey’s multiple comparisons test; ns: non-significant; mean ± SEM).

**Table 1 ijms-22-06036-t001:** Effects of in vivo and in vitro oocyte aging at 20 °C on the eyeing and hatching rates in common carp (mean ± SD). Means sharing a common alphabetical symbol do not differ significantly.

**In Vivo Oocyte Aging**														
**HPO**	**0**	**2**		**4**		**6**		**8**		**10**		**12**		**28**
Eyeing %	88 ± 9 ^a^	91 ± 5 ^a^		77 ± 16 ^ab^		66 ± 30 ^ab^		46 ± 25 ^bc^		15 ± 11 ^cd^		3 ± 4 ^d^		0 ± 0 ^d^
Hatching %	83 ± 16 ^a^	88 ± 10 ^a^		71 ± 21 ^a^		59 ± 31 ^ab^		31 ± 19 ^bc^		6 ± 5 ^c^		0.7 ± 1 ^c^		0 ± 0 ^c^
**In Vitro Oocyte Aging**														
**HPO**	**0**	**2**		**4**		**6**		**8**		**10**		**12**		**28**
Eyeing %	95 ± 4 ^a^		94 ± 3 ^a^		92 ± 3 ^a^		84 ± 8 ^a^		62 ± 6 ^b^		40 ± 5 ^c^		0 ± 0 ^d^	
Hatching %	94 ± 4 ^a^		92 ± 1 ^a^		88 ± 5 ^a^		68 ± 15 ^ab^		36 ± 22 ^bc^		21 ± 9 ^cd^		0 ± 0 ^d^	

HPO: Hours post-ovulation; HPS: Hours post-stripping.

## Data Availability

Data from the analysis are available from the corresponding author on request.

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
