# Peer review of "Histone Acetylation Dynamics during In Vivo and In Vitro Oocyte Aging in Common Carp Cyprinus carpio"

_ijms, 2021, doi:10.3390/ijms22116036_

Round 1

Reviewer 1 Report

In this research paper Waghmare et al. have outlined their observation about histone acetylation patterns during oocyte senescence of the common carp.

The manuscript is well organized and interesting. I have some suggestions to the authors that could improve their work:

  1. Please specify in the materials and method section how many independent experiments were performed?
  2. How was the isolated protein quantification executed?
  3. Were any loading control experiments performed?

Minor:

  1. There are some typo errors in the manuscript that could be avoided.

Author Response

In this research paper Waghmare et al. have outlined their observation about histone acetylation patterns during oocyte senescence of the common carp.

The manuscript is well organized and interesting. I have some suggestions to the authors that could improve their work:

A: We are glad that the reviewer found our manuscript as well-organized and interesting. We tried to improve the manuscript according to the reviewer’s suggestions and comments as explained below.

Please specify in the materials and method section how many independent experiments were performed?

A: We added an introductory paragraph indicating the number of experiments performed in the study in lines 291-293 of the revised manuscript.

How was the isolated protein quantification executed?

A: The protein quantification was determined by preloading the samples on 4–15% Criterion™ TGX™ Precast Midi Protein Gels (Bio-Rad, USA) followed by Coomassie staining. This sentence was mentioned in lines 334-336 of the original submitted manuscript.

Were any loading control experiments performed?

A: Yes, the loading control experiment was performed with anti-Histone H3 antibody (H0164-25UL; Sigma-Aldric). The western blot images of the loading control experiment were submitted along with the original submitted version. The normalization of the western blot images was performed at the data analysis step, using the corresponding stain-free gel image, and quantified using Image Lab 6.1.0 (Bio-Rad Laboratories, USA) software. Therefore, we skipped to mention the loading control experiment in the manuscript text.

Minor:

There are some typo errors in the manuscript that could be avoided.

A: We tried to correct the typo error in line 198 of the revised manuscript.

Reviewer 2 Report

Dear authors,

congratulations for the work. Please, check the attached document for some suggestions.

Thank you 

Best regards

Author Response

Comment

Answer

Reviewer #2:

This study reports the presence of acetylated Histone 3 (H3K9) and Histone 4 (H4K5, H4K8 and H4K12) in oocytes of carp (Cyprunys carpio) as well as their acetylating dynamic during in vivo and in vitro aging. In a time, when epigenetic modifications are explaining numerous biological processes, phenotypes and being reason of fertilization success/fail, works as the current one adds more knowledge to this field. Fish reproduction, though different to that of big mammals, represents an efficient resource of genetic material with work to.

In general terms, histone acetylation is not associated to oocyte aging in carps. Only, H4K12 is increased after 28 h of incubation in vitro. Whilst these results are contradictory to the most literature published up to date, as authors mentioned, this paradox could be due to species-specific reasons, highlighting, in addition, a well-developed protocol of oocyte conservation. However, a possibility that was not addressed in the manuscript is that maybe, carps display greater resistance to aging and longer incubation periods should be implemented. Alternatively, to trigger this process pharmacologically, if possible, may have been considered. Can the authors provide evidence that the biological processes that take place during oocyte-aging were reached during the incubation that they performed in the current experiment??

In this sense, an oocyte quality assessment (fluorescence microscopy, immunohistochemistry evaluations of proteins related with senescence, oxidative stress…) could have offered the confirmation of an oocyte quality that resembles that after aging.

In addition, since embryo production (in vitro and in vivo) directly depends on oocyte quality, such information would add more light to the results. Have the authors such kind of results (maybe previously published) to provide evidence of oocyte aging after 28 h of ovulation/stripping. If so, I strongly suggest including them in the document.

Apart from this, the paper is well written and well documented, with a recent and useful bibliography. In my opinion, the discussion is the best section of the manuscript. The main conclusion of the paper is well described. In addition, the information in regards of how the results agrees or disagrees with other similar work is nicely performed. Finally, I enjoyed with the hypothesis that the H4K12 increment could be due to a spontaneous activation of the oocytes.

To finish, please, check the paper attached below of this document to see more suggestions.

Summarizing the review, I do not find the current version of the manuscript ready to be published. Consider addressing the modifications suggested in this document and along the next pages for the improvement of the actual version.

Line 50: I stronggly suggest adding the species where such modifications have been reproted. This paragraph, to my mind, try to expose that epigenetic modifications are responsible of oocyte aging in several species and this could be also happening in fish. Whilst the extrapolation of this phenomenon seems appropiate to me, the authors should enphasize the extrapolation concept from other species to fishes.

Line 53: I suggest adding a linker as "similarly", for example, to emphasize that SIMILARLY than in other species, oocyte aging process could be occurring in fishes.

Line 72: Please, describe the name of this gene: Histone acetyltransferase 1?¿?¿

Line 74: Please, add that such processes remain unclear in fishes. That could help to link this paragph with the following one.

Line 80: Please, consider to remove these two sentences. Or at least, rewriting them in a way that emphasize that the current work based on western blot evaluation will support the works developed up to date based on inmunocitochemistry assays. To my mind, both techniques have strong beneffits and are more than accepted for the scientific community.

Line number 83: Consider to remove this sentence or reinforce the idea of studing oocyte aging i in fishes since they provide a big amount of high quality eggs reducing animal slaughter as happens in porcine, goats-sheep or cattle studies.

Line 85: Please, use this reference to reinforce concept mentioned in the previous note.

Line 102: In my opinion, starting the results with those showed in the supplementary file reduces the global quality perception of the readers. However, maybe this could be a personal opinion.

Line 106: I have my concerns with these two sentences:

Was the difference between the spots 80 and 113 statiscally significally p < 0.05?

1) If not, I strongly recommend removing this information from this section.

2) If so, please, rewrite the sentence properly indicating the significant between  both kind of samples:

- aged and non-aged oocytes (in vivo spot 80)

- aged and non-aged oocytes (in vitro spot 113).

Please, indicate which supplementary files show every single result.

Line 115: Please, add the Figure that shows this results.

Line 170: Please, add the meaning of 1, 2, 3, 4...

Line 190: Please, add the degrees.

Line 295: Please, describe how the authors calculated these rates.

Line 308: Please, add the comercial reference or if it was hadmade prepared, add a reference of a work where it has been used previously.

Line 310: Please, add the comercial reference or if it was hadmade prepared, add a reference of a work where it has been used previously.

We are glad that the reviewer found our study informative and our experimental animal, fish, as a proper organism for performing the study.

Once common carp oocytes are stored and getting aged, either in vivo or in vitro, they lose their fertilizing ability within 12-14 hours at 20 °C (the latest). We have performed such experiment with more than 30 individual females and published the results (Samarin et al., 2015, Samarin et al., 2019). Therefore, we would say that ageing of the common carp oocytes for 28 hours post ovulation and/or stripping is even two times more than the required time for totally losing the fertilizing ability (14 h). In addition, our unpublished results with grass carp Ctenopharhyngodon idella exhibited that only H4K12ac was modified during the oocyte ageing. We do believe hence that the other studied acetylation modifications except H4K12ac are not associated with the ageing process of the common carp oocytes. We would therefore say that examining the histone modifications at more advanced aged oocytes (more than 28 h) in the current study would not add more to our purpose which was studying these modifications up to the total loss of oocyte fertilizing ability caused by ageing (14 h in common carp).

The oocyte quality in this study was assessed by examining the eyeing and hatching rates (table 1). In our previously published papers however, we also examined the oxidative stress related enzymes, genes and oxidative products during oocyte ageing in common carp where we did not find any significant change up to 14 h (Samarin et al. 2019).

In our previously published papers, we provided the embryo mortality and larval malformation rates associated with oocyte ageing as well (Samarin et al., 2015, Samarin et al., 2019). In the current study, we present the hatching rates in table 1 which is referring to the number of arising embryos from different aged oocytes. What we are confident about anyway, is that after 14 hours of ageing oocytes in common carp, no embryo arises because the oocytes totally lose their fertilizing ability. Thus, there is no chance to provide any evidence about the embryos arising from more aged oocytes (after 14 h) as the reviewer suggested.

We are pleased that the reviewer found the discussion section and the conclusion part of our manuscript scientifically sound, and well-written.

We read the comments and suggestions of the reviewer in the text and tried to improve the manuscript according to them as are explaining below as well.

We hope that the revised version of the manuscript will be suitable for publication from the reviewer’s point of view.

.

We modified the sentences in lines 46-51 according to the reviewer’s suggestion.

We replaced the word “therefore” by “similarly” according to the reviewer’s suggestion in line 54 of the revised manuscript.

We agree with the reviewer’s suggestion and included the gene full name in the revised version of the manuscript in line 74.

We agree with the reviewer’s suggestion. We added the word “fish” in line 76.

We rewrote the sentence without comparing the techniques advantages over each other. The modified sentence is now presented in lines 81-82 of the revised manuscript.

To ease the study of oocyte aging, synchronous ovulation is very beneficial and is one of the prominent advantages of using common carp. We think that removing this sentence will somehow reduce the importance of common carp for using in this study. Therefore, we kept the sentence as it appeared in the original submitted version of the manuscript. We also added this sentence in lines 86-87 of the revised manuscript: “and there is no need to sacrifice the experimental animal as it might be required in other vertebrates”.

We tried to make this connection in lines 85-87 of the revised manuscript.

In general, we agree with the idea that the reviewer mentioned. However, in the current study we needed to present the results of the global histone modifications first and then later study the specific histone acetylation. Therefore, it was inevitable to follow the current order.

The differences between the spots 80 and 113 were not statically significant at p < 0.05. Therefore, we removed the related information from the result section according to the reviewer’s recommendation. We moved this part to the discussion section and placed it in lines 211-216 of the revised version of the manuscript.

We provide this information in lines 212 and 215.

We created the figure (1C) in which the comparison of in vivo and in vitro aged oocytes is presented. In addition, we modified the figure legend accordingly.

We added the description of the numbers in the line 178 of the revised version of the manuscript.

That was a typo error, we added the incubation temperature in line 198.

We described the calculations of eyeing and hatching rates in lines 311-315 of the revised manuscript.

The reference of the histone isolation procedure from where the homogenization buffer composition was adopted is given in line 325. The composition of the homogenization buffer is now included in the revised version of the manuscript in line 327.

The reference of the histone isolation procedure from where the salt wash buffer composition was adopted is given in the line 325. The composition of the salt wash buffer is now included in the revised version of the manuscript in line 329.